# Gait Asymmetry and Post-Traumatic Osteoarthritis Following Anterior Cruciate Ligament Rupture: A Preliminary Study

**DOI:** 10.3390/biology14020208

**Published:** 2025-02-16

**Authors:** Samuel Pringle, Kristiaan D’Août

**Affiliations:** Department of Musculoskeletal & Ageing Science, Institute of Life Course and Medical Sciences, University of Liverpool, Liverpool L7 8TX, UK; sam_pringle_1989@outlook.com

**Keywords:** post-traumatic osteoarthritis (PTOA), anterior cruciate ligament (ACL), gait asymmetry, duty factor, external knee adduction moment (EKAM), knee flexion moment (KFM)

## Abstract

Knee post-traumatic osteoarthritis (PTOA) is a type of osteoarthritis (OA), typically occurring in younger adults following serious joint injuries such as anterior cruciate ligament (ACL) tears. Walking gait asymmetry, whereby a person favours one leg over the other, leads to abnormal joint loading, which can drive the development of OA and PTOA and can cause chronic joint degeneration. Gait asymmetry-induced joint loading worsens late-stage OA severity; however, early-stage involvement and treatments, particularly in PTOA, are poorly defined. This research explored the involvement of gait asymmetry in early-stage PTOA following ACL tears in younger adults. Participants with historical ACL tears (ACL+) and participants with no previous joint trauma (ACL−) underwent gait analysis, determining gait asymmetry and abnormal joint loading severity. Our work found that gait asymmetry and joint load were greater in participants with previous ACL injuries, and gait asymmetry was seen to potentially cause increased knee flexion moments, a common joint load metric. Therefore, these preliminary data imply that gait asymmetry-induced joint loading may contribute to early-stage PTOA in subjects with ACL tears. This study was exploratory, and more research is required to validate and reinforce these findings before prospective treatments can be developed.

## 1. Introduction

Knee post-traumatic osteoarthritis (PTOA) is a common knee disease, causing approximately 12% of all symptomatic osteoarthritis (OA) [1,2] and 25% of knee OA cases [3], and results in significant pain and biomechanical dysfunction [4]. Function can be regained after surgical ACL reconstruction [5] but, unfortunately, the odds of PTOA development increase nearly 7-fold [6], and this occurs disproportionately in younger populations due to greater participation in high-intensity, dynamic activities [7,8]. Unfortunately, limited clinical interventions are available to prevent early PTOA development, with care pathways following major knee joint trauma drastically varying [9,10]. Further, the heterogeneity of OA and differing outcome measures and standards hinder the development of research and treatments [11,12,13]. With the global population expected to increase from 7.7 billion to 9.7 billion by 2050 [14], the impact of PTOA will only worsen. Therefore, understanding underlying factors that drive disease progression is paramount in developing preventative interventions.

Following ACL injury, patients often develop an asymmetrical walking gait, causing increased joint loading on the contralateral limb [15,16]. Notably, a 12-year study investigated late-stage disease severity and found that 80% unilateral knee OA patients developed bilateral knee OA through shifting joint load contralaterally to minimise stress on the ipsilateral joint [17]. This highlights the importance of walking gait asymmetry in joint loading. Conclusively, research investigating gait asymmetry-induced joint loading in early-stage PTOA progression is prudent to assist in the development of preventative care [9,10].

Duty factors represent the proportion of foot–ground contact time during gait, with duty factor asymmetry between legs quantifying greater ground contact time favouring one side to the other, providing a reliable gait asymmetry measure [18,19,20,21]. External knee adduction moment (EKAM) and knee flexion moment (KFM) are regarded as robust predictors of joint loading during the gait stance phase [22,23,24,25]. Importantly, EKAM and KFM are associated with knee OA progression in non-traumatic OA, particularly clinical outcomes such as bone marrow lesions and increased articular cartilage degeneration severity [22,24,26,27,28]. Further, joint moment–time integrals (i.e., angular impulse) provide increased accuracy as they consider the absolute magnitude of joint load during stance phase [22,23,24,29] and are predictive of articular cartilage loss over 12 months [30,31]. Importantly, one study linked KFM with increased medial and lateral cartilage degeneration following ACL rupture [32], with another identifying EKAM as a predictor of increased medial compartment contact forces in ALC rupture patients [33].

The purpose of this study was to investigate whether gait asymmetry is involved in early-knee PTOA progression following ACL rupture. We hypothesised that increased gait asymmetry could be implicated in the increased risk associated with knee PTOA progression in younger populations with ACL ruptures.

## 2. Materials and Methods

Participants constituted two groups, ACL− (*n* = 11) and ACL+ (*n* = 4), which evidenced no historical traumatic joint injuries or a complete ACL rupture (>12 months) with successful rehab completion, respectively (Table 1). Other inclusion criteria included being any sex and aged 25–50 years old. Exclusion criteria included a body mass index > 35, pregnancy, overt OA symptoms [34], and recent injuries or medical conditions affecting normal biomechanics.

**Gait biomechanics assessment:** Participants walked barefoot using their preferred pace and gait data were reported per leg for each participant across five trial runs. Sixty-seven infrared markers were attached to participants according to University of Liverpool lab standards. A 12 Oqus-7 infrared-camera motion capture system captured kinematics data (Qualisys, Gothenburg, Sweden; sampling rate—200 Hz) and Kistler force plates recorded ground reaction forces (GRF) (Kistler Inc., Winterthur, Switzerland; sampling rate—1000 Hz) with Qualisys Track Manager (QTM; Qualisys, Gothenburg, Sweden), collecting and synchronising datasets. The system was dynamically calibrated at the start of every experimental day, yielding standard errors relating to the spatial position of the markers below 1.5 mm. Force plates were reset between experiments. External joint moments were calculated based on inverse dynamics using Visual3D (C-Motion, Inc., Germantown, MD, USA) [35,36]. The measures included EKAM and KFM early (peak 1; EKAMp1 and KFMp1) and late (peak 2; EKAMp2 and KFMp2) stance peaks (Nm/kg) and their respective time integrals. The latter reflected the overall magnitude of the first and second halves of the stance phase (Nm·ms/kg; iEKAMp1, iEKAMp2, iKFMp1 and iKFMp2) (Figure 1). All external joint moment data were normalised in relation to body mass (kg). All stance and swing times for each limb were extracted using Visual 3D to establish mean duty factor values (stance time/(stance time + swing time)) for each leg [18]. Duty factor asymmetry was calculated using the following formula:DF asymmetry = 100 (1 − (mean duty factor right/mean duty factor left)) 

This provided a percentage deviation from zero, whereby zero percent is absolute symmetry, and positive values indicate a longer DF for the right limb.

**Statistical analysis:** Independent unpaired *t*-tests were used to compare differences in all functional variable means between ACL+ and ACL− groups and Shapiro–Wilk tests confirmed whether data were normally distributed. Linear regression modelling was applied to analyse predictive relationships between independent and dependent variables, with each data point constituting one trial metric. All statistical analyses were conducted using Prism v9.4.1.

## 3. Results

Fifteen participants completed the experiments and we observed a 78% difference in duty factor asymmetry between group sample means (*p* = 0.0067). Additionally, the following functional variables each demonstrated significant mean differences: EKAMp1 (26%; *p* = 0.0119, ACL+: 0.54 Nm/kg, ACL−: 0.41 Nm/kg), EKAMp2 (49%; *p* = 0.0361, ACL+: 0.49 Nm/kg, ACL−: 0.30 Nm/kg), iEKAMp1 (30%; *p* = 0.0366, ACL+: 109.39 Nm·ms/kg, ACL−: 80.84 Nm·ms/kg), KFMp1 (37%; *p* = 0.0117, ACL+: 1.02 Nm/kg, ACL−: 0.70 Nm/kg), iKFMp1 (44%; *p* = 0.0178, ACL+: 190.4 Nm·ms/kg, ACL−: 121.00 Nm·ms/kg), and iKFMp2 (60%; *p* = 0.0329, ACL+: 83.47 Nm·ms/kg, ACL−: 44.92 Nm·ms/kg). However, no significant mean differences were found between iEKAMp2 (58%; *p* = 0.0892, ACL+: 99.03 Nm·ms/kg, ACL−: 54.75 Nm·ms/kg) and KFMp2 (39%; *p* = 0.1392, ACL+: 0.47 Nm/kg, ACL−: 0.31 Nm/kg) (Figure 2). When assessing the linear relationship between duty factor asymmetry (independent variable) and each functional variable (dependent variable), KFMp2 (R^2^ = 0.665; *p* = < 0.001) and iKFMp2 (R^2^ = 0.505); *p* = < 0.001) registered moderate associations with duty factor asymmetry in the ACL+ group. All other variables, across both ACL+ and ACL− groups, demonstrated either nominal R^2^ values or no significant results (Figure 3).

## 4. Discussion

This exploratory study provided supporting evidence that increased gait asymmetry-induced joint loading may be involved in knee PTOA progression in subjects with ACL ruptures. ACL+ participants demonstrated greater gait asymmetry relative to ACL− participants and displayed heightened joint loading across several knee PTOA risk-factors (Figure 2). Further, the ACL+ cohort displayed moderate associations between gait asymmetry and KFMp2 and iKFMp2 in the second half of the stance phase (Figure 3).

**Gait asymmetry:** The study revealed that participants with historical ACL ruptures (
$\bar{x}$
 = 3%) exhibited a 78% increase in duty factor asymmetry compared to ACL− participants (
$\bar{x}$
 = 1.3%; Figure 2). Supporting this, a study spanning 24 months (*n* = 40) demonstrated increased gait asymmetry prevalence in ACL patients following reconstruction [37]. Further, another project illustrated 1.6% and 1% gait asymmetry sample mean reductions associated with reduced knee pain symptoms in knee OA patients [38], validating the clinical relevance of the group sample mean differences. 

**External knee adduction moment:** The study highlighted significantly heightened differences in both EKAM peaks that favoured the ACL+ group, with EKAMp1 (26%) and EKAMp2 (49%). Additionally, the time integral iEKAMp1 exhibited a significant 30% increase in the ACL+ group, with iEKAMp2 also showing an increase at 58%, but this did not reach significance (Figure 2). These data augment current knowledge illustrating greater EKAM magnitude in ACL patients [39] or subjects at high risk of ACL rupture [40,41]. Additionally, Miyazaki et al. (2002) [42] established an association of ~25% increase in EKAM peak magnitude with 6.6 times heightened risk of radiographic knee OA progression over 6 years, highlighting the relevance of the group mean differences in this study. Further, a greater EKAM presence increases compressive forces through the patellofemoral and tibiofemoral joint compartments [22,43], while a 12-year study established increased joint load as a major OA progression contributor [17]. Therefore, ACL rupture subjects may exhibit increased joint moment magnitude, increasing susceptibility to PTOA development.

**Knee flexion moment:** The evidence indicated KFM peaks and time integrals display greater prevalence of joint load in ACL+ participants. The significant differences were as follows: KFMp1—37%; iKFMp1—44%; iKFMp2—60% (Figure 2). Supporting this, evidence has been cited of increased KFM peaks and time integrals in earlier disease stages (30–55 years old), contributing to increased progressive cartilage loss [24,27,32]. These heightened knee flexion moments increase knee extensor force, generating a greater patellofemoral and tibiofemoral joint reaction force [24,25,44]. This suggests that ACL+ participants experience higher patellofemoral and tibiofemoral joint loading over longer periods, a major risk factor in PTOA progression [26].

**Gait asymmetry in post-traumatic osteoarthritis:** The finding of the relationship between duty factor asymmetry, as a potential predictor, and OA markers (KFMp2 (R^2^ = 0.665) and iKFMp2 (R^2^ = 0.505)) is to our knowledge a novel finding (Figure 3). As such, no direct supporting evidence exists; however, indirect insight is available. Notably, gait asymmetry is significantly increased in younger patients with ACL ruptures compared to uninjured controls [45,46], which can ultimately increase joint load, resulting in greater risk of cartilage matrix degradation [17]. Further, two studies demonstrated increasing OA severity as KFMp2 and iKFMp2 worsened [24,47]. Therefore, the findings associating duty factor asymmetry with KFMp2 and iKFMp2 hold relevance. Combined with the increased prevalence of KFM and EKAM in the ACL+ participants, there exists additional support for the hypothesis: increased gait asymmetry-induced joint loading may be involved in the increased risk associated with early-knee PTOA progression in younger populations with ACL ruptures. Rehabilitation programmes after ACL reconstruction can help to restore symmetry [48,49].

**Limitations and future directions:** Caution should be exercised when drawing definitive interpretations. We used small sample sizes. To estimate the impact of this, we performed a posteriori power analyses (two-tailed, α = 0.05) in G*Power 3.1.9.7 for two main outcomes: duty factor asymmetry (=ACL−: 0313 ± 0.00892, *n* = 11; ACL+: 0.0298 ± 0.00875, *n* = 4) and EKAMp1 (ACL− 0.714 ± 0.198, I = 11; ACL 1.066 ± 0.199, I = 4). The effects sizes were 1.89 and 1.77, respectively. The calculated powers were 0.849 and 0.801, respectively. This means that our power for these important outcomes is good, despite the low numbers. However, the lack of females in the ACL+ group means it may not accurately represent the overall population, and we propose increasing samples sizes [50] and using proportional representation [11] in future studies, especially when ultimately aiming to make clinical recommendations. Women are at a 3–5 times higher risk of ACL injuries, 50–90% of which progress to OA [51], and so any such recommendations can be especially beneficial for women. Using G*Power, the recommendation would be to expand the research to approximately 40–50 participants (allowing a 20% drop-out rate) for a power of 0.95 at α = 0.05 [52]. Additionally, study designs including radiographic imaging (X-Ray, MRI, CT) or detailed physical examination [53] to quantify cartilage degradation and gait asymmetry rectification plans with follow-ups, would provide tangible insights on the effectiveness of managing gait asymmetry as a preventative treatment. Further, another deliberation is whether the ACL rupture is the main cause of PTOA [51,54,55], or whether the secondary trauma of the reconstruction itself is a significant contributor [56,57]. Equally, how, and when does gait asymmetry begin impacting PTOA? Is it simply a by-product of the ACL injury and PTOA development? Or does it have greater implications prior to PTOA in the causation of the ACL rupture, ultimately being the underlying driving factor? These are important considerations whose investigation would greatly help the broader view around the topic in terms of understanding the root causes of the problem in PTOA onset and progression.

## 5. Conclusions

The findings of this study indicate that gait asymmetry is increased in participants with historical ACL ruptures. Mechanical loading (KFM and EKAM) is heightened in ACL participants relative to participants with no historical knee joint trauma. Further, gait asymmetry may increase knee flexion moment magnitude, suggesting the involvement of gait asymmetry in knee PTOA progression in ACL participants. Future studies should expand the study design to validate and build on these preliminary findings.

## Figures and Tables

**Figure 1 biology-14-00208-f001:**
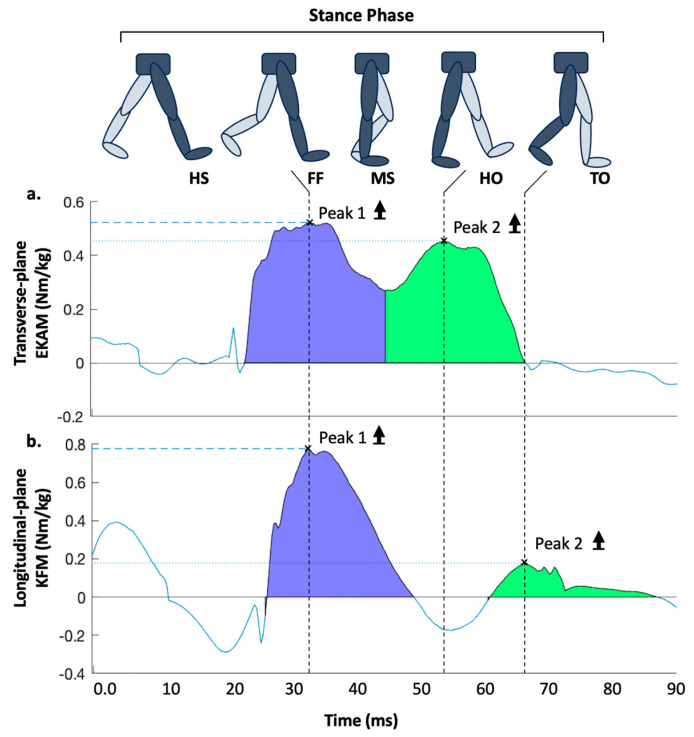
Distribution curves for (**a**) EKAM (frontal plane) and (**b**) KFM (sagittal plane) knee joint moments (Nm/kg) amid the walking stance phase, taken from a single-sample trial. Positive and negative values signify (**a**) adduction and abduction and (**b**) knee flexion and extension moments, respectively. Each moment curve presents two calculated peaks, which are signified by crosses. Peak 1 (blue) and Peak 2 (green) represent the first and second halve of the stance phase, respectively. The time integral for each phase (1 and 2) of the respective curves (EKAM and KFM) were computed relative to time (Nm·ms/kg) when the knee moments (Nm/kg) were positive. **Abbreviations:** HS, heel-strike; FF, foot-flat; MS, mid-stance; HO, heel-off; TO, toe-off.

**Figure 2 biology-14-00208-f002:**
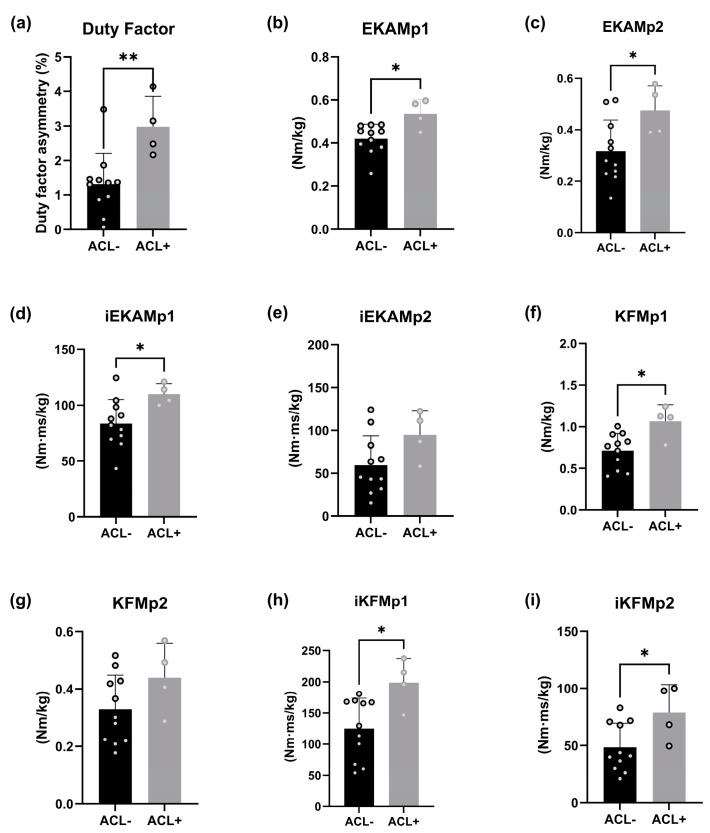
The sample mean distribution of ACL+ and ACL− groups for (**a**) gait asymmetry as a product of duty factor asymmetry (stance time/(stance time + swing time)), (**b**) EKAMp1 and (**c**) EKAMp2, (**d**) iEKAMp1 and (**e**) iEKAMp2, (**f**) KFMp1 and (**g**) KFMp2, and (**h**) iKFMp1 and (**i**) iKFMp2.* = *p* < 0.05, ** = *p* < 0.01 observed following am independent unpaired two-sample *t*-test.

**Figure 3 biology-14-00208-f003:**
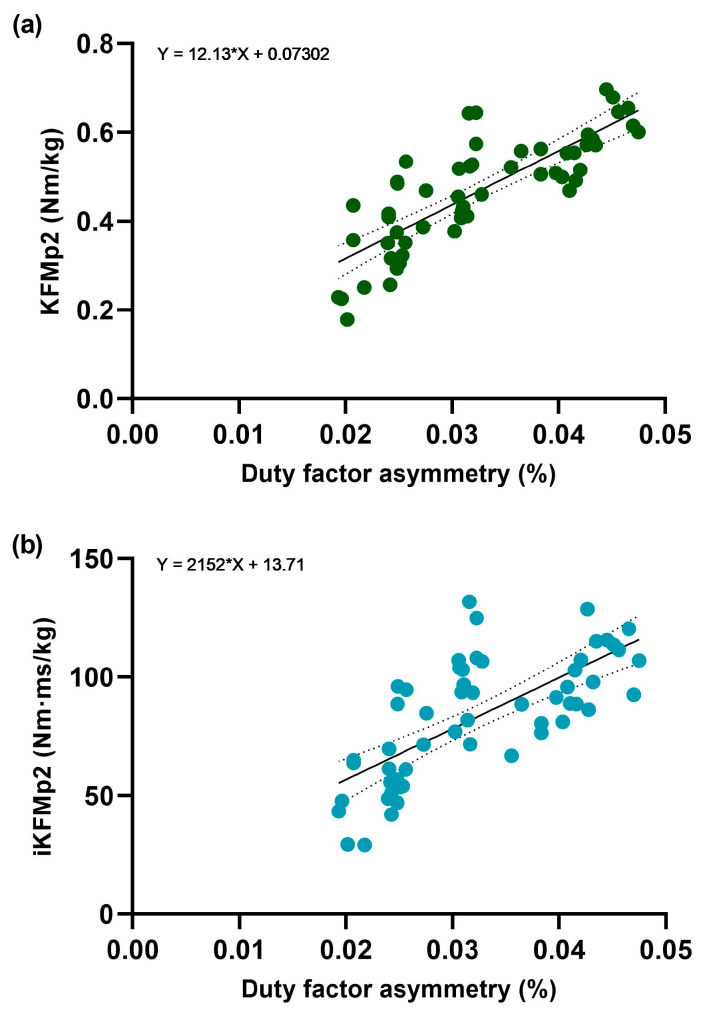
A linear regression model displaying predictive relationships between the independent variable (x), gait asymmetry as a product of duty factor asymmetry (stance time/(stance time + swing time)) and independent variables (y) for (**a**) KFMp2 (R^2^ = 0.665; *p* = < 0.001) and (**b**) iKFMp2 (R^2^ = 0.505); *p* = < 0.001) in the ACL+ group.

**Table 1 biology-14-00208-t001:** Participant demographic variables, with group 1 consisting of participants with no articular cartilage ligament rupture (ACL−) and group 2 (ACL+) consists of participants with historical ACL ruptures.

Demographic Variable	Group 1. ACL−	Group 2. ACL+	*p* Value
Sex (male/female) (*n*)	5/6	4/0	
Age (*y*) *	35 ± 6	32 ± 1	0.1158
Mass (kg) *	71.3 ± 12.5	86.6 ± 7.9	0.0998
Height (cm) *^†^	174 ± 11.3	185 ± 3.9	<0.001
Body mass index (kg/m^2^) *	23 ± 2.3	25.2 ± 2.0	0.1442
Years post-surgery	-	6.0 ± 3.2	-

* values are mean ± SD; ^†^ significant differences observed following two-sample *t*-test between ACL− and ACL+ groups at α = 0.05 and confidence intervals = 95%.

## Data Availability

Data can be made available upon reasonable request.

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
