# Peer review of "Gait Asymmetry and Post-Traumatic Osteoarthritis Following Anterior Cruciate Ligament Rupture: A Preliminary Study"

_biology, 2025, doi:10.3390/biology14020208_

Round 1

Reviewer 1 Report (Previous Reviewer 4)

Comments and Suggestions for Authors This cross-sectional study compared gait asymmetry parameters between patients with or without anterior cruciate ligament (ACL) tears and found that patients with ACL injuries suffered altered gait biomechanics which might contribute to post-traumatic OA. My humble opinion is that the research question tested is too obvious, and the ACL group only have 4 patients, which is too little even for a pilot study. This limitation has been addressed in the discussion, but I hope the authors can demonstrate the effects and power of the current analysis.  

Minor: The authors said they recruited 11 ACL- patients but in Table 1, the group has 6 men and women.

Author Response

Reviewer 1

Comment 1:

This cross-sectional study compared gait asymmetry parameters between patients with or without anterior cruciate ligament (ACL) tears and found that patients with ACL injuries suffered altered gait biomechanics which might contribute to post-traumatic OA. My humble opinion is that the research question tested is too obvious, and the ACL group only have 4 patients, which is too little even for a pilot study. This limitation has been addressed in the discussion, but I hope the authors can demonstrate the effects and power of the current analysis.  

Response:

Thank you and it is an excellent idea to provide power analyses. We have performed a posteriori power analyses (two-tailed, a=0.05) in G*Power for two main outcomes: duty factor asymmetry (NACL: 0313 ± 0.00892, n=11; ACL: 0.0298 ± 0.00875, n=4) and EKAMp1 (NACL 0.714 ± 0.198, n=11; ACL 1.066 ± 0.199, n=4).

Effects sizes are 1.89 and 1.77, respectively.

Calculated powers are 0.849 and 0.801, respectively.

We conclude that, despite the small sample size for the ACL group, we have good power. We have added these results to the manuscript and feel this is a real improvement – thank you very much for this comment.

Comment 2:

Minor: The authors said they recruited 11 ACL- patients but in Table 1, the group has 6 men and women.

Response:

Thank you for spotting this error. The nACL group consists of 5 male and 6 female (total n=11). We have corrected the table.

Reviewer 2 Report (New Reviewer)

Comments and Suggestions for Authors

Dear Authors,

Thank you for the opportunity to review your manuscript. The topic is highly relevant and addresses a significant gap in understanding the mechanisms of post-traumatic osteoarthritis (PTOA) development in younger adults following ACL injury. While the manuscript is promising, there are several areas where improvements are necessary before it can be considered for publication.

General Comments:

·         Please submit a clean copy of the manuscript without any tracked changes or comments, as this will facilitate a clearer and more focused review process.

·         There are discrepancies in terminology, specifically terms like "gait asymmetry-induced joint loading" versus "gait asymmetry and joint load." Please standardize the terminology throughout the manuscript to ensure clarity and consistency.

Methods:

·         The limited sample size, especially the absence of female participants in the ACL+ group, could significantly impact the generalizability of your findings. Could you provide insight into the challenges faced during recruitment or the rationale behind the chosen demographic?

·         A detailed description of the calibration procedures for the motion-capture system is necessary. Clarify how you ensured the accuracy of the joint moment calculations, which are critical for validating your study's findings.

·         Elaborate on how duty factor asymmetry was calculated and discuss its relevance in the context of gait analysis. This will enhance understanding for readers who may be unfamiliar with this measure.

Results:

·         There are noted differences in body mass, height, and BMI between the nACL and ACL groups, which, despite not being statistically significant, could influence the results. Consider conducting subgroup analyses to see if outcomes vary by age, gender, or time since injury. This approach could provide deeper insights and allow for more personalized conclusions.

Discussion:

·         Discuss how variations in rehabilitation quality and duration might affect gait asymmetry and joint loading outcomes. It’s crucial to understand these potential confounders to interpret the results accurately.

·         Explore how interventions targeting gait asymmetry could potentially mitigate the risk of developing PTOA. This discussion could significantly contribute to the practical implications of your findings.

·         Please elaborate on the potential gender-specific effects in PTOA development post-ACL injury. Considering the gender disparities in your sample, this discussion could add valuable depth to your study.

Author Response

Reviewer 2

Comment 1:

Thank you for the opportunity to review your manuscript. The topic is highly relevant and addresses a significant gap in understanding the mechanisms of post-traumatic osteoarthritis (PTOA) development in younger adults following ACL injury. While the manuscript is promising, there are several areas where improvements are necessary before it can be considered for publication.

General Comments:

Please submit a clean copy of the manuscript without any tracked changes or comments, as this will facilitate a clearer and more focused review process.

Response:

Thank you, we have provided a clean version. Reviewers wanting to see changes can use the “compare documents” version in Word.

Comment 2:

There are discrepancies in terminology, specifically terms like "gait asymmetry-induced joint loading" versus "gait asymmetry and joint load." Please standardize the terminology throughout the manuscript to ensure clarity and consistency.

Response:

Thank you, we have now checked the manuscript for consistency. However we distinguish between the terms mentioned. “Gait asymmetry-induced joint loading” is used where imply a causal relationship between the two. Where we observe both increased asymmetry and joint load, but cannot draw a causal relationship (or direction) we prefer to refer to the more cautious “gait asymmetry and joint load”.

Comment 3:

Methods:

The limited sample size, especially the absence of female participants in the ACL+ group, could significantly impact the generalizability of your findings. Could you provide insight into the challenges faced during recruitment or the rationale behind the chosen demographic?

Response:

Response:

We have added some a posteriori power analyses in the Discussion, and elaborated on sex differences, to help alleviate this concern (we realise this technique must be used with caution). Even if this shows good power (>0.8) we fully acknowledge that out sample, esp. for the ACL group, is small and consists of only one sex. This work was part of a postgraduate research project which meant we had limited scope to recruit larger and sex/gender-balanced groups. Even though ACL injuries are more common in women than in men, we have not been able to recruit women for the nACL group. In a larger follow-up study, we must address this.

Comment 4:

A detailed description of the calibration procedures for the motion-capture system is necessary. Clarify how you ensured the accuracy of the joint moment calculations, which are critical for validating your study's findings.

Response:

Thank you and this is a good point. We calibrated the Qualisys system at the start of every experimental day. Calibration consisted of a dynamic spatial calibration, which in our lab we accept if the standard error of the measurements is below 1.5 mm (it is typically 0.7 – 0.8 mm) which we argue is below skin motion artefacts and therefore sufficient for whole-body gait. Definition of the lab coordination system was using a manufacturer-provided L-frame positioned exactly at corner of our force plate matrix (we have four plates): for the moment calculations it is indeed crucial to not only have accurate force magnitudes but also a correct location of the centre of force (as well as knee marker positions). The Kistler force plates were reset between experiments to remove any possible drift. We have added some detail in the Methods section to make this clear to the reader.

Comment 5:

Elaborate on how duty factor asymmetry was calculated and discuss its relevance in the context of gait analysis. This will enhance understanding for readers who may be unfamiliar with this measure.

Response:

Duty Factor (DF) asymmetry was calculated as per the equation in the manuscript. It is the ratio of the right and to left DF, subtracted from 1 and multiplied by 100 to yield a percentage. This means that full symmetry yields a value of zero and positive values mean a longer DF in the left leg. We have clarified this in the text and amended the equation because it was confusing. Thank you very much for this comment.

Comment 6:

Results:

There are noted differences in body mass, height, and BMI between the nACL and ACL groups, which, despite not being statistically significant, could influence the results. Consider conducting subgroup analyses to see if outcomes vary by age, gender, or time since injury. This approach could provide deeper insights and allow for more personalized conclusions.

Response:

This is a justified comment and we agree this could influence the results. Unfortunately, we believe we do not have sufficient data to further stratify our groups with confidence. We will take this valid comment forward to any follow-up study.

Comment 7:

Discussion:

Discuss how variations in rehabilitation quality and duration might affect gait asymmetry and joint loading outcomes. It’s crucial to understand these potential confounders to interpret the results accurately.

Response:

Thank you and this is and excellent point. We have no data on rehabilitation of our subjects so can only speculate that these factors may play a role in our data. Speaking more generally, this factor has been acknowledged and we have referenced some articles in the Discussion (Ebert et al., 2018; Hadizadeh et al., 2016) outlining functional and gait asymmetries post ACL surgery/rehab.

Comment 8:

Explore how interventions targeting gait asymmetry could potentially mitigate the risk of developing PTOA. This discussion could significantly contribute to the practical implications of your findings.

Response:

Thank you, and ultimately we would like to be able to offer such clinical recommendations. However, a larger study would be needed to justify this, based on experimental data. At this moment we can only speculate because we cannot draw a causal relationship between asymmetry and OA in our data.

Comment 9:

Please elaborate on the potential gender-specific effects in PTOA development post-ACL injury. Considering the gender disparities in your sample, this discussion could add valuable depth to your study.

Response:

Thank you and this is a very fair point. We only have men in our ACL sample, so we cannot quantify such effects. We can speculate that, given the 3-5 times higher risk of ACL injuries and the 50-90% progression to OA (Wang et al., 2020, Arthr Res Ther), any normalisation of gait, including making it symmetrical to distribute loads, can be especially beneficial for women. We have added this to our Discussion.

Reviewer 3 Report (New Reviewer)

Comments and Suggestions for Authors

General characteristics and evaluation of the reviewed article: Gait Asymmetry and Post-Traumatic Osteoarthritis Following Anterior Cruciate Ligament Rupture: A Preliminary Study

The article investigates the role of gait asymmetry in the early development of post-traumatic osteoarthritis (PTOA) following anterior cruciate ligament (ACL) rupture. By focusing on younger populations, the study addresses a critical gap in understanding how biomechanical factors, particularly gait asymmetry, contribute to early-stage PTOA, which accounts for 12% of all symptomatic osteoarthritis cases.

Using motion-capture technology and force plates, the authors measured gait asymmetry (duty factor) and joint loading parameters in participants with a history of ACL rupture (ACL+, n = 4) and controls without prior joint trauma (ACL-, n = 11). The study found significant differences in duty factor asymmetry (78%) and joint loading measures, such as the external knee adduction and flexion moments, with asymmetry correlating strongly with certain joint loading metrics (R² = 0.665 for KFMp2).

Despite these promising findings, the study's small sample size limits the generalizability of the results. Additionally, the lack of direct cartilage assessment restricts conclusions regarding the structural impact of gait asymmetry on joint health. Longitudinal studies with larger cohorts and imaging-based evaluations are needed to confirm the causal relationship between altered biomechanics and PTOA progression.

The article is interesting, addresses a timely and important topic and definitely fits the scope of the journal. It is written generally correctly and requires only minor corrections and additions before further processing and acceptance for publication. Below are my points and detailed comments.

Minor comments:

The introduction is far too short and does not present the problem in sufficient detail.

Expanding the discussion of osteoarthritis in the first paragraph could significantly enhance the introduction by highlighting the importance and relevance of this condition. The prevalence of osteoarthritis is influenced by various factors, including occupational activities, sports participation, musculoskeletal injuries, obesity, and gender. Incorporating detailed information about these factors, supported by relevant literature, would provide a robust foundation for the topic. The following references are recommended for inclusion in this section:

https://doi.org/10.3390/healthcare12161648

DOI: 10.1056/NEJMcp1903768

Please expand the introduction with more detailed information on ACL injuries, typical mechanism of injury and add the latest literature. I recommend adding the following literature items to the passage indicated: DOI 10.1177/23259671241275663 ; DOI 10.1088/1742-6596/1736/1/012027 ;  DOI 10.1016/j.humov.2024.103298 ;

The study was conducted on a small group of participants (ACL+: 4, ACL-: 11), which limits the generalizability of the results and their statistical power. In order to increase the statistical power of the analysis and improve the representativeness of the results, it is suggested to expand the group of participants to at least 40-50 people, with gender and age proportions. Please describe this limitation in more detail and outline plans for further research.

Despite the indication of the potential impact of gait asymmetry on PTOA, the lack of use of imaging techniques (e.g., MRI) limits the ability to confirm the link between biomechanics and cartilage damage.  Incorporating techniques such as MRI or PET/CT to assess cartilage status could directly confirm the effect of gait asymmetry on structural damage to the joint. Please describe this aspect in the discussion by referring to the paper: DOI 10.3390/app9194102

The ACL+ group consisted only of men, which does not allow generalizing the results to the entire population. The inclusion of women and a wider range of demographics will allow for a more complete analysis of the results and their application to different population groups. Please describe this limitation more fully and provide solutions in future studies.

The inclusion of women and a broader demographic range will allow for a more complete analysis of the results and their application to different population groups. The authors should consider designing a longitudinal study to track biomechanical changes and their impact on PTOA development over the long term.

The study should consider the effect of rehabilitation on gait asymmetry and evaluate the effectiveness of corrective strategies in minimizing the risk of PTOA.

This research highlights the potential role of gait asymmetry in early PTOA development, paving the way for future studies to explore targeted interventions aimed at reducing biomechanical risks in post-ACL injury populations.

I congratulate the authors on an interesting study and wish them further success !

Author Response

Reviewer 3

General characteristics and evaluation of the reviewed article: Gait Asymmetry and Post-Traumatic Osteoarthritis Following Anterior Cruciate Ligament Rupture: A Preliminary Study

The article investigates the role of gait asymmetry in the early development of post-traumatic osteoarthritis (PTOA) following anterior cruciate ligament (ACL) rupture. By focusing on younger populations, the study addresses a critical gap in understanding how biomechanical factors, particularly gait asymmetry, contribute to early-stage PTOA, which accounts for 12% of all symptomatic osteoarthritis cases.

Using motion-capture technology and force plates, the authors measured gait asymmetry (duty factor) and joint loading parameters in participants with a history of ACL rupture (ACL+, n = 4) and controls without prior joint trauma (ACL-, n = 11). The study found significant differences in duty factor asymmetry (78%) and joint loading measures, such as the external knee adduction and flexion moments, with asymmetry correlating strongly with certain joint loading metrics (R² = 0.665 for KFMp2).

Despite these promising findings, the study's small sample size limits the generalizability of the results. Additionally, the lack of direct cartilage assessment restricts conclusions regarding the structural impact of gait asymmetry on joint health. Longitudinal studies with larger cohorts and imaging-based evaluations are needed to confirm the causal relationship between altered biomechanics and PTOA progression.

The article is interesting, addresses a timely and important topic and definitely fits the scope of the journal. It is written generally correctly and requires only minor corrections and additions before further processing and acceptance for publication. Below are my points and detailed comments.

Minor comments:

Comment 1:

The introduction is far too short and does not present the problem in sufficient detail.

Response:

Thank you, and we acknowledge the Introduction is brief. This manuscript is a Brief Report and at 2926 words, we are close to the journal’s guidance word count of 3000 maximum. For the revised version, we have added some details requested by the reviewers, so we have very limited scope to make the Introduction substantially longer. Therefore we have not extended the manuscript, but it now contains new references, including some (e.g. Sharma et al., 2021; Wang et al., 2020) which provide such more extensive background. We especially thank the reviewer for suggesting references in their other comments, which is very helpful, and we have added these where most relevant.

Comment 2:

Expanding the discussion of osteoarthritis in the first paragraph could significantly enhance the introduction by highlighting the importance and relevance of this condition. The prevalence of osteoarthritis is influenced by various factors, including occupational activities, sports participation, musculoskeletal injuries, obesity, and gender. Incorporating detailed information about these factors, supported by relevant literature, would provide a robust foundation for the topic. The following references are recommended for inclusion in this section:

https://doi.org/10.3390/healthcare12161648

DOI: 10.1056/NEJMcp1903768

Response:

Thank you for this suggestion. We have added Sharma et al (2021). The other reference is about cartilage frictional properties and repair approaches and, while interesting as such, we feel it is less relevant for inclusion in this brief report.

Comment 3:

Please expand the introduction with more detailed information on ACL injuries, typical mechanism of injury and add the latest literature. I recommend adding the following literature items to the passage indicated: DOI 10.1177/23259671241275663 ; DOI 10.1088/1742-6596/1736/1/012027 ;  DOI 10.1016/j.humov.2024.103298 ;

Response:

Thank you for this suggestion. We have now referenced Mercurio et al (2024). The other papers are on diagnostic accuracy and hamstring fatigue, respectively, and, while interesting, we feel they would distract from the core topic of this manuscript, a brief report only.

Comment 4:

The study was conducted on a small group of participants (ACL+: 4, ACL-: 11), which limits the generalizability of the results and their statistical power. In order to increase the statistical power of the analysis and improve the representativeness of the results, it is suggested to expand the group of participants to at least 40-50 people, with gender and age proportions. Please describe this limitation in more detail and outline plans for further research.

Response:

Thank you -we have now done a posteriori power calculations on two key outcomes (DF asymmetry and EKAMp1) and extended the relevant section in the Discussion.

Comment 5:

Despite the indication of the potential impact of gait asymmetry on PTOA, the lack of use of imaging techniques (e.g., MRI) limits the ability to confirm the link between biomechanics and cartilage damage.  Incorporating techniques such as MRI or PET/CT to assess cartilage status could directly confirm the effect of gait asymmetry on structural damage to the joint. Please describe this aspect in the discussion by referring to the paper: DOI 10.3390/app9194102

Response:

Thank you, this is a very valid comment. We have no imaging data but have now elaborated on this issue in our Discussion and used the reference provided. We found this reference very interesting, showing that detailed physical examination can be superior to MRI assessment.

Comment 6:

The ACL+ group consisted only of men, which does not allow generalizing the results to the entire population. The inclusion of women and a wider range of demographics will allow for a more complete analysis of the results and their application to different population groups. Please describe this limitation more fully and provide solutions in future studies.

The inclusion of women and a broader demographic range will allow for a more complete analysis of the results and their application to different population groups. The authors should consider designing a longitudinal study to track biomechanical changes and their impact on PTOA development over the long term.

Response:

We fully agree and have made this point very clear in the manuscript. A larger follow-up study should have a good gender/sex balance and be more diverse and representative in general.

Comment 7:

The study should consider the effect of rehabilitation on gait asymmetry and evaluate the effectiveness of corrective strategies in minimizing the risk of PTOA.

Response

Thank you and this is an excellent point. We have no data on rehabilitation of our subjects so can only speculate that these factors may play a role in our data. Speaking more generally, this factor has been acknowledged and we have referenced some articles (Ebert et al., 2018; Hadizadeh et al., 2016) on the restoration of symmetry after ACL reconstruction.

Comment 8:

This research highlights the potential role of gait asymmetry in early PTOA development, paving the way for future studies to explore targeted interventions aimed at reducing biomechanical risks in post-ACL injury populations.

I congratulate the authors on an interesting study and wish them further success !

Response:

Thank you very much for these encouraging words. We hope that this paper, with the improvements based on your and the other reviewers’ comments, will be acceptable for publication and stimulate further research on this important topic.

Round 2

Reviewer 1 Report (Previous Reviewer 4)

Comments and Suggestions for Authors

Duplicated legends for Figure 2.

Otherwise, I am happy with the corrections. 

Reviewer 2 Report (New Reviewer)

Comments and Suggestions for Authors

Thank you for revising your manuscript. The current version is satisfactory. 

This manuscript is a resubmission of an earlier submission. The following is a list of the peer review reports and author responses from that submission.

Round 1

Reviewer 1 Report

Comments and Suggestions for Authors

I appreciated the topic focused on evaluating the relationship between gait asymmetry-induced loading and knee OA progression. The manuscript is very well written and the data is clearly presented. Please see my specific comments below regarding suggestions to strengthen the interpretation of the results.

Specific issues that need to be addressed by author(s):

1) It would be beneficial to provide the reader with a demographic breakdown of the participants in the study for each group (i.e., ACL- and ACL+). For example age group, sex, body mass index, race, time since ACL rupture. Especially given the hypothesis specifically references the younger population. Additionally, the authors refer to early-stage PTOA development throughout the manuscript. However, participants were admitted to the study if ACL rupture took place >12 months with no specific data on time since injury (i.e., how can one know the stage of OA development for each participant if the time since injury is not provided?).        

2) In Figure 2, inclusion of the individual data points for each bar graph would help in the interpretation of the data. 

3) The authors provide linear regression data for KFMp2 and iKFMp2 since moderate associations were found with duty factor asymmetry. The linear analysis of the remaining variables, regardless of significance, should be presented as well given that the manuscript is limited on data.   

Author Response

Reviewer 1

I appreciated the topic focused on evaluating the relationship between gait asymmetry-induced loading and knee OA progression. The manuscript is very well written and the data is clearly presented. Please see my specific comments below regarding suggestions to strengthen the interpretation of the results.

Specific issues that need to be addressed by author(s):

Comment 1:
It would be beneficial to provide the reader with a demographic breakdown of the participants in the study for each group (i.e., ACL- and ACL+). For example age group, sex, body mass index, race, time since ACL rupture. Especially given the hypothesis specifically references the younger population. Additionally, the authors refer to early-stage PTOA development throughout the manuscript. However, participants were admitted to the study if ACL rupture took place >12 months with no specific data on time since injury (i.e., how can one know the stage of OA development for each participant if the time since injury is not provided?).        

Response:
Thank you, and this is an excellent point. We have now added a table (Table 1) with those data.

Comment 2:
In Figure 2, inclusion of the individual data points for each bar graph would help in the interpretation of the data. 

Response:
Thank you. We have now added the individual data points to the plots in Figure 2.

Comment 3:
The authors provide linear regression data for KFMp2 and iKFMp2 since moderate associations were found with duty factor asymmetry. The linear analysis of the remaining variables, regardless of significance, should be presented as well given that the manuscript is limited on data.   

Response:
Thank you. We understand this concern. However, because this is a short preliminary report we wanted to keep it as concise and focussed as possible. We also wanted to avoid “fishing” for results by analysing more data than needed to test our hypotheses.

Reviewer 2 Report

Comments and Suggestions for Authors

The main aim of this paper ‘to investigate whether gait asymmetry-induced joint loading is involved in early-knee PTOA progression following ACL rupture’ is very relevant and important. 

However, this study does not address this research question. The authors relate asymmetry in stance time with knee loading (KAM and KFM). They do not investigate cartilage quality, so cannot make any conclusions regarding OA progression/development at al.

Furthermore there are many incorrect interpretations of existing literature and important methodological shortcomings (low sample size with inappropriate statistics).

Simple summary

Line 14: What do you mean with ‘master’s research’?

Abstract

Line 29: I suggest to search for an alternative for the term ‘duty factor asymmetry’ as this a difficult, non-scientifically term. Why not using ‘gait asymmetry’?

Line 32: You specifically mention that KAM is an external moment (EKAM). For KFM you do not specify whether it is internal or external. Be consistent.

Line 37: This conclusion is too short sighted, even if you increase the sample size of this study. The differences observed between both groups do not implicate that these differences lead to OA progression in the ACL group. For such statements you need to follow-up the cartilage status as well.

Introduction

The introduction is too short-sighted and contains several mistakes regarding the existing literature, with several papers incorrectly referred to.

Line 48: you state that up to 87% of the ACL patients will develop PTOA. This percentage seems very high. Most literature states that about 50% of the patients develop PTOA. The paper of Friel et al., where you refer to, is a review paper, not an epidemiological study or systematic review. Please revise this with better references.

Line 57: I do not agree that ACL patients show increased joint load. Several studies showed reduced external knee flexion moments compared to healthy controls, even years after their ACL injury (eg. Lepley et al. 2018, J athletic training; Kaur et al. 2016, Sports Med). The 2 studies were the authors refer to (Wellsandt and Erhart-Hledik) do not support the statement of the authors at all. First, these 2 studies do not include a control group, they only compare between limb differences and evolution over time. So you can not concluded whether the joint moments are increased or decreased compared to what is ‘normal’ (this requires control group). Second, these studies show lower joint moments in the involved leg compared to the uninvolved leg. So there is no indication of overloading the involved leg at all.

Line 73: these references refer to non-traumatic OA. At least mention this, because the reader should be aware of this. The mechanism of over/underloading might be totally different between different fenotypes of OA. In Non-traumatic OA, it could be that overloading the medial compartment lead to faster progression, but it is speculated that this is different in ACL patients. One can expect for example underloading in the beginning of the rehabilitation, with cartilage deconditioning and then relative overloading when the patient gets back to normal activities. Furthermore altered intra-articular kinematics can lead to shifts in loading areas of the contact surface of the joint.

Line 74: Joint moment time integrals: suggest to use the correct biomechanical term for this ‘impulse’.

Line 81: you cannot investigate OA progression if you do not involve measures/markers of it (eg. cartilage quality or quantity observed on MRI, OA classification on Rx..)

Material and methods

The authors only recruited 4 ACL patients. This is – even for a preliminary analyses – very limited. One of the inclusion criteria is that the ACL injury need to be at least 12 months ago. Is there an upper limit? This makes the very small ACL sample, also very heterogenous.

Furthermore it is not clear whether both groups are matched?

Statistics: the authors use parametric tests. For such small sample they need to use non-parametric tests or just descriptive statistics.

Results

The authors refer to longitudinal and transversal plane. Please use the standard biomechanical terminology for this: sagittal plane for flexion/extension, frontal plane for abduction/adduction.

Author Response

Reviewer 2

Comment 1:
The main aim of this paper ‘to investigate whether gait asymmetry-induced joint loading is involved in early-knee PTOA progression following ACL rupture’ is very relevant and important. 
However, this study does not address this research question. The authors relate asymmetry in stance time with knee loading (KAM and KFM). They do not investigate cartilage quality, so cannot make any conclusions regarding OA progression/development at al.

Response:
Thank you for this important and justified comment. We have not directly measured cartilage quality or thickness and have now acknowledged our lack of data on cartilage degradation clearly in the Discussion as a limitation. We focus entirely on changes post ACL rupture, which we know is correlated with OA progression in general, as are our outcomes such as EKAM. We acknowledge we do not have cartilage data for our specific subjects. A future, larger study should definitely include cartilage measurements, and we than the reviewer for this comment.

Comment 2:
Furthermore there are many incorrect interpretations of existing literature and important methodological shortcomings (low sample size with inappropriate statistics).

Response:
Thank you for this comment. We have attempted to use the literature correctly, but would very much appreciate concrete examples where we failed to do that so we could further improve our manuscript. In terms of the sample size and statistics, we acknowledge that we have low numbers, which is why we label it as a Preliminary Study. Any future study will involve larger numbers, and we have used the current data to do a power analysis for such study. We believe one benefit of this paper is that it providing input data for power calculations. This will help design future and more expensive (i.e. including radiography) studies.

Comment 3:
Simple summary
Line 14: What do you mean with ‘master’s research’?

Response:
Thank you and this is a good comment. This work was part of a Masters dissertation but that is not relevant and we have removed the reference to it.

Comment 4:
Abstract
Line 29: I suggest to search for an alternative for the term ‘duty factor asymmetry’ as this a difficult, non-scientifically term. Why not using ‘gait asymmetry’?

Response:
Thank you, however, “duty factor” is a very common scientific term in biomechanics, and we have defined it in the Introduction (line 67). We have now also made this clearer in the Abstract. Asymmetry in duty factor is a very commonly used variable in clinical biomechanics. Sometimes, “stance duration” is used which is the same as duty factor (but not normalised to stride duration), see e.g. the textbook “Dynamics of Human Gait” (Vaughan, Davis and O’Connor,2nd edition, 1999, p. 10).

Comment 5:
Line 32: You specifically mention that KAM is an external moment (EKAM). For KFM you do not specify whether it is internal or external. Be consistent.

Response:
Thank you, and we agree. We have now specified that our knee flexor moment is also an external moment (line 32).

Comment 6:
Line 37: This conclusion is too short sighted, even if you increase the sample size of this study. The differences observed between both groups do not implicate that these differences lead to OA progression in the ACL group. For such statements you need to follow-up the cartilage status as well.

Response:
Thank you and this relates to Comment 1. We agree, and we have amended the concluding sentence of the Abstract.

Comment 7:
Introduction
The introduction is too short-sighted and contains several mistakes regarding the existing literature, with several papers incorrectly referred to.

Response:
Please see our response to Comment 2.

Comment 8:
Line 48: you state that up to 87% of the ACL patients will develop PTOA. This percentage seems very high. Most literature states that about 50% of the patients develop PTOA. The paper of Friel et al., where you refer to, is a review paper, not an epidemiological study or systematic review. Please revise this with better references.

Response:
We have now updated this reference with a recent systematic review and meta-analysis (Webster and Hewett, 2022, Clin J Sport Med).

Comment 9:
Line 57: I do not agree that ACL patients show increased joint load. Several studies showed reduced external knee flexion moments compared to healthy controls, even years after their ACL injury (eg. Lepley et al. 2018, J athletic training; Kaur et al. 2016, Sports Med). The 2 studies were the authors refer to (Wellsandt and Erhart-Hledik) do not support the statement of the authors at all. First, these 2 studies do not include a control group, they only compare between limb differences and evolution over time. So you can not concluded whether the joint moments are increased or decreased compared to what is ‘normal’ (this requires control group). Second, these studies show lower joint moments in the involved leg compared to the uninvolved leg. So there is no indication of overloading the involved leg at all.

Response:
Thank you and we apologise for this mistake – joint load as measured by joint moments are indeed shown to lower (the compressive contact forces may be increased though), even though we found increased loads (but that is a result and should not be part of the Introduction). We have removed this section to focus purely on our topic, which is gait asymmetry, and amended the whole manuscript where necessary.

Comment 10:
Line 73: these references refer to non-traumatic OA. At least mention this, because the reader should be aware of this. The mechanism of over/underloading might be totally different between different fenotypes of OA. In Non-traumatic OA, it could be that overloading the medial compartment lead to faster progression, but it is speculated that this is different in ACL patients. One can expect for example underloading in the beginning of the rehabilitation, with cartilage deconditioning and then relative overloading when the patient gets back to normal activities. Furthermore altered intra-articular kinematics can lead to shifts in loading areas of the contact surface of the joint.

Response:
Thank you, this is an excellent point and we now state this applies to non-traumatic OA.

Comment 11:
Line 74: Joint moment time integrals: suggest to use the correct biomechanical term for this ‘impulse’.

Response:
Mechanically, impulse is the integral of force (not moment) over time. The moment-time integral is the “angular impulse” which we have now mentioned. However, we believe this term may not be familiar to many readers, and thus prefer to also keep the very descriptive and correct term “moment-time integral”.

Comment 12:
Line 81: you cannot investigate OA progression if you do not involve measures/markers of it (eg. cartilage quality or quantity observed on MRI, OA classification on Rx..)

Response:
We agree (see also Comment 1) and have added some wording (incl. in the abstract and discussion) accordingly. Unfortunately we do not have this data (at least for this preliminary study) so have to limit ourselves to relationships with ACL injury (we now provide more details on this in a new table, Table 1) but not cartilage degradation.

Comment 13:
Material and methods
The authors only recruited 4 ACL patients. This is – even for a preliminary analyses – very limited. One of the inclusion criteria is that the ACL injury need to be at least 12 months ago. Is there an upper limit? This makes the very small ACL sample, also very heterogenous.

Response:

Thank you, we have now provided ACL history data in the new Table 1. Our subjects were 3-10 years post-surgery.

Comment 14:
Furthermore it is not clear whether both groups are matched?

Response:
Thank you, for this study it was impossible to match our groups perfectly, but they are of similar age and BMI as shown in the new Table 1. The major mis-match is in sex distribution, which we have acknowledged as a limitation in our Discussion.

Comment 15:
Statistics: the authors use parametric tests. For such small sample they need to use non-parametric tests or just descriptive statistics.

Response:
We agree our data set is small and we need to be careful with interpretation of statistical results, which we have mentioned in the Discussion. We have tested for normality and as a result used parametric tests. However we have now also added individual data points to the main outcome plots (Figure 2) which will help the reader interpret the data.

Comment 16:
Results
The authors refer to longitudinal and transversal plane. Please use the standard biomechanical terminology for this: sagittal plane for flexion/extension, frontal plane for abduction/adduction.

Response:
Thank you and we fully agree, we have corrected the terminology.

Reviewer 3 Report

Comments and Suggestions for Authors

For Abstract and Simple Summary:

Here are some suggestions to improve the text, structured to align with academic and scientific standards:

1.Contextualize the Research Problem: Begin by offering a broader context of PTOA's impact on public health, emphasizing its prevalence, economic burden, and the quality-of-life implications for affected individuals. This approach can underscore the significance of addressing gait asymmetry as a potential contributory factor.

2.Define Key Concepts Early: Ensure that key terms and concepts (e.g., PTOA, ACL, gait asymmetry, joint loading) are clearly defined at their first mention. This will aid readers unfamiliar with specific terminology to grasp the study's focus without needing external references.

3.Literature Review: Expand the literature review to include a more comprehensive analysis of existing research. Highlight gaps in knowledge regarding the onset and progression of early-stage PTOA, especially in the context of ACL injuries. This could involve a more detailed examination of previous studies that have identified gait asymmetry and abnormal joint loading as critical factors in PTOA development and progression.

4.Rationale for the Study: Strengthen the justification for your research. While the summary mentions the exploration of gait asymmetry in early-stage PTOA, elucidate why understanding this aspect is crucial for developing preventive strategies or interventions. Clarify how your study addresses a gap in the existing literature or offers a novel perspective on the issue.

5.Objectives and Hypotheses: Clearly articulate the specific objectives or hypotheses your study aims to test. This clarity will help readers understand the direction of your research and the specific questions you seek to answer. It’s beneficial to explicitly state these objectives or hypotheses towards the end of the introduction to guide readers into the methodology section.

6.Methodological Preview: Offer a brief preview of your methodological approach. While detailed methods belong in a later section, a concise statement about the study design, participant selection, and analytical techniques can set the stage for the detailed exposition to come.

7.Potential Implications: Briefly hint at the potential implications of your findings. Understanding early-stage PTOA's mechanistic underpinnings could inform more targeted rehabilitation strategies, enhance preventive measures, or guide future research directions. This preview can motivate readers to delve deeper into your findings.

For Introduction:

1.Streamline and Clarify the Presentation of Statistics and References: While the inclusion of statistics and references is essential for establishing the context and significance of your research, ensure that each piece of data directly supports your narrative. Consider integrating statistics more seamlessly into the text to improve readability and flow. For example, instead of listing statistics and references in succession, weave them into a cohesive narrative that clearly articulates the problem and its significance.

2.Enhance the Explanation of Key Concepts: Early in the introduction, define crucial terms such as "PTOA," "ACL rupture," "gait asymmetry," and "joint loading." While these terms are likely familiar to your target audience, a concise definition or description can ensure clarity and enhance the introduction's inclusivity for interdisciplinary readers.

3.Expand on the Rationale for the Study: While you've identified the importance of studying gait asymmetry in PTOA development, further elaboration on why this specific focus (e.g., gait asymmetry-induced joint loading) could contribute uniquely to the field would strengthen the introduction. Discuss the gap in current knowledge or research that your study aims to fill, highlighting the potential impact of your findings on the development of preventative strategies or interventions.

4.Clarify the Link Between ACL Ruptures and PTOA: Provide a more detailed explanation of the mechanisms through which ACL ruptures lead to increased PTOA risk, emphasizing the role of altered biomechanics and joint loading patterns. This detail will help readers understand the biological and mechanical underpinnings of your research hypothesis.

5.Discuss the Potential Impact of Global Population Trends: Briefly elaborate on how the expected increase in the global population could exacerbate the public health challenge posed by PTOA, particularly in the context of younger populations and high-intensity activities. This connection will reinforce the urgency and relevance of your research.

6.Introduce the Study's Objectives and Hypotheses More Clearly: While you mention the purpose and hypothesis of your study towards the end of the introduction, making this statement more prominent and detailed could enhance the reader's understanding of your research aims. Consider framing the study's objectives and hypotheses in a way that directly addresses the identified research gap, linking them explicitly to the anticipated contribution of your work to the field.

7.Technical Precision and Terminology: Ensure technical terms and abbreviations are used consistently and correctly throughout. For instance, check for typographical errors such as "ALC rupture" which should presumably be "ACL rupture." Maintaining precision in terminology is crucial for scientific accuracy and credibility.

For Materials and Methods:

1.Clarify Participant Selection Criteria: While the inclusion and exclusion criteria are generally clear, providing rationale for specific choices (e.g., age range, BMI limit) would offer readers insight into how these criteria might influence the study's outcomes or its applicability to a broader population. For example, explaining why participants with a body mass index (BMI) over 35 were excluded could clarify potential impacts of obesity on gait biomechanics and OA risk.

2.Detail Regarding Rehabilitation Success: For ACL+ participants, specify what criteria were used to define "successful rehab completion." This could include functional benchmarks, clinical assessments, or patient-reported outcomes. Clarifying this could help in understanding the homogeneity of the participant pool regarding their post-injury recovery status.

3.Methodological Precision: While the description of the gait biomechanics assessment procedure is thorough, including more details about the protocol for attaching infrared markers and defining joint axes could enhance reproducibility. For instance, detailing the anatomical landmarks used for marker placement or the criteria for defining segment coordinate systems would be beneficial.

4.Statistical Methods: The use of independent unpaired t-tests and Shapiro-Wilk tests is appropriate for the analyses described. However, elaborating on the choice of these tests and any assumptions underlying their use (such as the assumption of normality for t-tests) would strengthen the methodological rigor of the study. Additionally, when discussing linear regression modeling, specifying the criteria for model selection, the handling of potential confounders, and the assessment of model fit would provide a more complete picture of the analytical approach.

5.Figure Description Clarity: The description of Figure 1 is detailed but could benefit from simplification and clarification. Instead of embedding the explanation of what positive and negative values signify within the figure description, consider introducing these concepts earlier in the text where the measures (EKAM and KFM) are first explained. This would make the figure description more concise and focused on what the reader should observe in the figure.

6.Enhance Explanation of Duty Factor Asymmetry Calculation: The formula for calculating duty factor asymmetry is provided, but a brief explanation of how this calculation reflects gait asymmetry could aid understanding. For instance, elaborating on why a deviation from zero percent symmetry is significant in the context of gait analysis would be helpful.

7.Technical and Formatting Consistency: Ensure consistency in the formatting of abbreviations, units of measurement, and statistical notations throughout the section. For example, maintaining a consistent format for all measurements (Nm/kg, Nm·ms/kg) and statistical software versions (Prism v9.4.1) enhances the professional presentation of the manuscript.

For Results:

1.Clarify Statistical Significance and Presentation of Results: While the presentation of statistical outcomes (e.g., P-values, percentage differences) is clear, ensuring consistent formatting for P-values (e.g., P < 0.05, not P = < 0.05) and providing a brief interpretation of what these differences mean in a clinical or biomechanical context would enhance readability and applicability.

2.Detail on Non-significant Results: For variables where no significant differences were observed, a brief discussion on the potential implications or reasons why these differences might not have been significant could provide valuable insights. This could include considerations of sample size, variability in participant rehabilitation outcomes, or the sensitivity of the measures used.

3.Interpretation of R² Values: The section reports R² values to describe the strength of linear relationships, but including a brief explanation of what these values imply about the relationship between duty factor asymmetry and joint loading in the ACL+ group would be beneficial. Clarifying how these associations might influence clinical outcomes or rehabilitation strategies could make the results more informative to readers.

4.Enhancement of Figure Descriptions: While Figures 2 and 3 are referenced, ensuring that figure captions are sufficiently detailed for readers to understand what the figures depict without needing to refer back to the text is crucial. Additionally, consider explaining the significance of the asterisks (* and **) in Figure 2 within the figure caption or the main text to clarify the levels of statistical significance represented.

5.Discussion of Moderate Associations: The identification of moderate associations warrants a brief discussion within the results section, even if a more comprehensive analysis is reserved for the discussion section. Offering insights into why certain variables show moderate associations with duty factor asymmetry could provide immediate context for the findings.

6.Correction of Typographical and Formatting Errors: Ensure that typographical errors (e.g., "P = < 0.001" should be "P < 0.001") are corrected and that there is consistency in the presentation of numerical and statistical data. This includes uniform use of decimal places and consistent formatting of statistical indicators.

7.Accessibility of Statistical Data: While detailed statistical data is provided, summarizing key findings in a way that highlights their clinical or practical significance could make the results more accessible to readers not specialized in statistical analysis. For instance, discussing the potential impact of observed differences in gait asymmetry and joint loading on rehabilitation practices or patient outcomes could enhance the section's relevance.

For Discussion:

1.Critical Analysis and Contextualization of Findings: While the discussion highlights the study's main findings, deeper analysis comparing these results with existing literature would strengthen the narrative. For each key finding, discuss how it aligns or diverges from previous studies and theorize potential reasons for these similarities or differences. This approach not only contextualizes your results within the broader field but also invites further investigation.

2.Clarify Novel Contributions: Explicitly state the novel contributions of your study early in the discussion. While you mention the novel finding regarding duty factor asymmetry, emphasizing the uniqueness of your contributions (e.g., the specific relationships between gait asymmetry and PTOA progression markers) early on helps to highlight the significance of your work.

3.Address Limitations More Broadly: You've outlined specific limitations related to sample size and gender representation. Expanding this section to include discussions on methodological constraints (e.g., the use of specific gait analysis tools, the definition of successful rehabilitation) and their potential impact on the study's findings would provide a more comprehensive view of the study's scope and applicability.

4.Future Directions with Broader Implications: While recommendations for future research are provided, elaborating on how these future studies could impact clinical practice or patient outcomes would be beneficial. For instance, discuss potential preventative strategies or interventions for PTOA that could be developed based on understanding gait asymmetry's role in its progression.

5.Interdisciplinary Insights: Consider integrating insights from related fields (e.g., physiotherapy, orthopedics, sports science) to discuss the implications of your findings. This multidisciplinary approach can broaden the appeal of your study and suggest avenues for collaborative research.

6.Statistical Significance vs. Clinical Relevance: Distinguish between statistical significance and clinical relevance, particularly in the context of observed differences in joint loading and gait asymmetry. Discuss how these differences might translate into clinically meaningful outcomes or interventions.

7.Enhanced Conclusion: Conclude the discussion with a strong, concise statement summarizing the study's key findings, its contributions to the field, and the potential implications for patient care and future research. This summary should encapsulate the essence of your work and its significance within the broader research landscape.

For Conclusions:

1.Strengthen the Link Between Findings and Clinical Implications: While the conclusions highlight the key findings, elaborating on the clinical implications of increased gait asymmetry and mechanical loading in ACL-ruptured individuals would add value. Explain how these findings could influence rehabilitation strategies, preventive measures, or the design of interventions to mitigate PTOA progression.

2.Specify the Nature of the Mechanical Loading: The terms "KFM" and "EKAM" are mentioned as being heightened in ACL participants, but the conclusion would benefit from a brief explanation of these terms for clarity and to ensure it is accessible to a broader audience. Additionally, specifying how these forms of mechanical loading contribute to the risk of PTOA could provide a more comprehensive understanding of the study's outcomes.

3.Highlight the Novelty and Contribution of the Study: Emphasize any novel contributions your study makes to the existing body of knowledge on PTOA and ACL rupture. If your findings offer new insights into the mechanisms of PTOA progression or the role of gait asymmetry, stating this explicitly will underscore the significance of your work.

4.Recommendations for Future Research: While you suggest expanding study designs for future research, providing more specific directions based on your findings could be beneficial. For example, recommend investigating the effectiveness of specific interventions aimed at reducing gait asymmetry or the long-term impact of altered mechanical loading on joint health in ACL-ruptured individuals. This guidance can help shape the focus of subsequent studies in this area.

5.Acknowledge Limitations: Briefly acknowledge any major limitations of the current study within the conclusions. This transparency can provide context for the findings and recommendations for future research, offering a more balanced view of the study's contributions and areas for further investigation.

6.Call to Action for Clinicians and Researchers: Conclude with a strong call to action, encouraging clinicians and researchers to consider the implications of gait asymmetry and mechanical loading in the management and study of PTOA in ACL-ruptured patients. This can foster a more integrated approach to addressing this common and challenging clinical issue.

Comments on the Quality of English Language

1.Grammar and Spelling Corrections:

•In the phrase "anterior cruciate liga- ment (ACL) tears," ensure that "ligament" is not hyphenated incorrectly due to line breaks.

•"Walking gait asymmetry, whereby a person favours one leg over the other, leads to increased and abnormal joint loading..." - Consistency in American vs. British English should be maintained. If the journal prefers American English, consider changing "favours" to "favors."

•Correct the typo "ALC rupture patients" to "ACL rupture patients" in the section discussing EKAM as a predictor.

2.Language Precision and Academic Tone:

•Consider using more precise language that reflects the research's exploratory nature without asserting causality prematurely. For example, "Our work found that gait asymmetry and joint load were greater in participants with previous ACL injuries..." could be revised to "The findings suggest that participants with previous ACL injuries exhibited greater gait asymmetry and joint load..."

•The phrase "This master’s research explored the involvement of gait asymmetry..." might be better phrased as "This study explored the role of gait asymmetry in the early stages of PTOA following ACL tears, focusing on younger adults."

3.Consistency in Terminology:

•Ensure consistent use of terms and abbreviations throughout the document. For instance, if "post-traumatic osteoarthritis" is initially introduced as "PTOA," continue to use the abbreviation consistently thereafter.

•When referring to "knee flexion moment" and "external knee adduction moment," ensure that these terms are used consistently throughout the text without abbreviating for the first instance in each section.

4.Clarity and Readability:

•Break complex sentences into shorter, more digestible ones to improve readability. This is particularly useful in sections with dense information, such as the Methods or Results.

•Use bullet points or numbered lists to present the inclusion and exclusion criteria in the Materials and Methods section for better clarity.

5.Technical Details and Formatting:

•Verify the accuracy of all technical terms, abbreviations, and the reference numbering system to ensure they align with standard conventions in the field.

•Ensure that all figures and tables are correctly cited in the text and that their descriptions are clear and informative.

6.References and Citations:

•Check the formatting of references to ensure consistency with the chosen citation style of the journal (e.g., APA, Harvard, etc.).

•Verify that hyperlinks (e.g., DOI links) are active and correct.

7.Ethical Statements and Acknowledgments:

•Review the ethical statements and acknowledgments to ensure they meet the journal's requirements, including clear statements of informed consent and data availability.

Author Response

Reviewer 3

Response:
This review seems to have been AI-generated and we are ignoring it, after discussion with the journal’s editorial team.

Reviewer 4 Report

Comments and Suggestions for Authors

In this brief report, the gait asymmetry parameters between subjects with ACL rupture and those without. The novelty lies with the comprehensive assessment of infrared-camera motion-capture and Kistler force plates. The limitations of the pilot study have been addressed by the authors in the discussion.

Major comments:

I would appreciate it if the definition of the key parameters (iEKAMp1, iEKAMp2, iKFMp1 and iKFMp2) be defined and attached as supplementary material. It would help the understanding of readers new to this field.

Minor comments:

Please remove the master’s research from the simple summary

Comments on the Quality of English Language

No comment

Author Response

Reviewer 4

Comment 1:
In this brief report, the gait asymmetry parameters between subjects with ACL rupture and those without. The novelty lies with the comprehensive assessment of infrared-camera motion-capture and Kistler force plates. The limitations of the pilot study have been addressed by the authors in the discussion.
Major comments:
I would appreciate it if the definition of the key parameters (iEKAMp1, iEKAMp2, iKFMp1 and iKFMp2) be defined and attached as supplementary material. It would help the understanding of readers new to this field.

Response:
Thank you. For the sake of brevity, we believe adding Supplementary Materials for this purpose might not be necessary. We provide references for the EKAM and other variables (standard terms used in this field) which less familiar readers can refer to. We believe this will resolve any issues.

Comment 2:
Minor comments:
Please remove the master’s research from the simple summary

Response:
Thank you, we agree. We have now removed this.